# Surfactant-Free Synthesis of Three-Dimensional Perovskite Titania-Based Micron-Scale Motifs Used as Catalytic Supports for the Methanol Oxidation Reaction

**DOI:** 10.3390/molecules26040909

**Published:** 2021-02-09

**Authors:** Nathaniel Hurley, Luyao Li, Christopher Koenigsmann, Stanislaus S. Wong

**Affiliations:** 1Department of Chemistry, State University of New York at Stony Brook, Stony Brook, NY 11794-3400, USA; nathaniel.hurley@stonybrook.edu (N.H.); LuyaoSBU@hotmail.com (L.L.); 2Department of Chemistry, Fordham University, Bronx, NY 10458, USA; ckoenigsmann@fordham.edu

**Keywords:** platinum nanoparticles, metal oxide, synthesis, electrocatalysis, interfacial chemistry

## Abstract

We synthesized and subsequently rationalized the formation of a series of 3D hierarchical metal oxide spherical motifs. Specifically, we varied the chemical composition within a family of ATiO_3_ (wherein “A” = Ca, Sr, and Ba) perovskites, using a two-step, surfactant-free synthesis procedure to generate structures with average diameters of ~3 microns. In terms of demonstrating the practicality of these perovskite materials, we have explored their use as supports for the methanol oxidation reaction (MOR) as a function of their size, morphology, and chemical composition. The MOR activity of our target systems was found to increase with decreasing ionic radius of the “A” site cation, in order of Pt/CaTiO_3_ (CTO) > Pt/SrTiO_3_ (STO) > Pt/BaTiO_3_ (BTO). With respect to morphology, we observed an MOR enhancement of our 3D spherical motifs, as compared with either ultra-small or cubic control samples. Moreover, the Pt/CTO sample yielded not only improved mass and specific activity values but also a greater stability and durability, as compared with both commercial TiO_2_ nanoparticle standards and precursor TiO_2_ templates.

## 1. Introduction

Ferroelectric perovskite-type oxides, with a general formula of ABO_3_, are noteworthy for their advantageous dielectric, piezoelectric, electrostrictive, and electro-optic properties [1,2,3,4,5,6,7]. Understanding the behavior of and the preparation of this interesting family of metal oxide materials with structure-dependent physical properties, at the nanoscale, are of importance to developing viable applications. Specifically, titanate-based perovskites (i.e., B = Ti) have found practical utility within the context of medical applications [8], photocatalysis [7,9,10], and energy storage [11,12]. In addition, they have been used as catalyst supports for electrochemical reactions [13]. Therefore, ensuring a systematic understanding of the growth and properties of the materials with respect to different morphologies and chemical compositions represents an important and achievable scientific goal.

Within these ABO_3_ systems, “A” represents a large cation, whereas “B” is a smaller cation that coordinates onto oxygen. Specifically, we report herein on the synthesis of a series of 3D alkaline earth-metal perovskites ATiO_3_ (A = Ca, Sr, and Ba) with uniform micron-sized spherical morphology, using a TiO_2_-template directed method. From an electrochemical and electrocatalytic perspective, we note that 3D nano-/micro-hierarchical spheres combine the merits of nanometer-sized building blocks (i.e., shortened diffusion distances and high electrode/electrolyte contact surface areas), with the benefits of either micrometer- or sub-micrometer-sized assemblies (i.e., thermodynamic stability and high tap density) [14,15].

The target materials were chosen, because many perovskite materials are electronically conductive, possess reasonable proton transport properties, and maintain chemically functionalizable outer surfaces, thereby rendering them as promising candidates for support materials within highly acidic environments, typically utilized by conventional small molecule oxidation [16]. Previous work has centered on the use of ABO_3_ perovskites (wherein “A” = Sr, La, Ba, or Ca; “B” = Ti, Ni, Fe, or Ru) as potential metal oxide support materials in the context of the methanol oxidation reaction (MOR). In particular, it has been suggested that the use of a complex metal oxide material as a support for Pt may not only lower the overpotential for MOR but also facilitate the complete oxidation of methanol to products, such as CO_2_ [17,18].

In relevant prior efforts from our group, we reported on the connection between synthesis control and MOR performance with SrRuO_3_ (SRO) [19]. In particular, we determined a correlation between morphology and the MOR performance of faceted SRO octahedra yielding improved performance metrics, as compared with rounded nanoparticles. These observed enhancements were attributed to the increased surface area of the faceted octahedra (i.e., 11.43 m^2^/g) vs. that of their rounded particulate counterparts (i.e., 2.86 m^2^/g). This increase in surface area likely led to a corresponding increase in active sites available for reactions [19]. Hence, this study demonstrated that morphology and shape clearly mattered for electrocatalysis.

Nevertheless, chemical composition was also a leading indicator of performance. In this context, we had synthesized various metal oxides of different sizes (i.e., ~35 to 150 nm), including TiO_2_, RuO_2_, SrTiO_3_, and SrRuO_3_ as supports for Pt nanoparticles (NPs) [13]. Our results, incorporating supporting mechanistic studies, showed that the SrRuO_3_ substrate with immobilized Pt NPs at its surface evinced the best MOR performance, as compared with all of the other substrate materials tested. In effect, the presence of Ru within SrRuO_3_ not only contributed to a significant increase in MOR activity (manifested as higher steady state current densities) but also resulted in an overall shift to lower MOR onset potentials. Significantly, we concluded that chemical composition, as opposed to the size of the support, was the more significant determinant of the onset potential, corroborating the idea that our perovskite materials could adsorb hydroxyl groups at their external surfaces, to facilitate MOR.

Therefore, to build upon these prior efforts, based on the importance of morphology, shape, and chemical composition, we focused herein on generating titania-based perovskites, characterized by different chemical compositions. Hence, the purpose and novelty of this paper are to (i) detail the perovskite growth conditions/mechanism for 3D ATiO_3_ materials; (ii) obtain data and report on the MOR activity of 3D ATiO_3_, when used as supports for Pt; and (iii) rationalize the improvement in MOR activity, as compared with commercial TiO_2_ standards.

Another significant consideration relates to the synthesis protocol used to generate the perovskites themselves. Specifically, the synthetic route can determine not only size, shape, and morphology but also the achievable active surface area available to the perovskite catalysts [12,13,20,21]. A number of fabrication methods have been proposed over the years, to ensure chemical purity and structural integrity, including but not limited to hydrothermal and solvothermal syntheses [20,21], solid state reactions [22,23], sonochemical protocols [24], and arc-melting methods [25]. However, some of these procedures, especially those related to solid state reactions, sonochemistry, and arc-melting, tend to be limited by reproducibility, yield little if any morphological control, and/or are more prone to cost and technical limitations. Therefore, in terms of additional novelty, we emphasize that the synthesis method implemented herein consists of a surfactant-free, two-step hydrothermal/sol–gel calcination process.

In the current report, this procedure was modified from a previously reported protocol that had centered on the use of a Lutensol ON50 surfactant [26]. The spheres produced by this Lutensol ON50–based method measured about 1 micron in average size, but these were smaller than the 3–6 micron diameter particles routinely generated herein. Specifically, in the method described in this study, the first step involved the hydrothermal synthesis of micron-scale particles of TiO_2_, which subsequently served as the template building blocks for further reactivity. These particles were then dispersed in ethanol and mixed with the relevant hydroxide salt, containing either Ca^2+^, Sr^2+^, or Ba^2+^ ions, respectively. Given the porous nature of the as-prepared three-dimensional, hierarchical TiO_2_ motif, a relatively uniform distribution of the salts could be deposited onto and within the spheres after physical sonication, thereby forming a “sol–gel” precursor. The TiO_2_ templates were sufficiently robust to withstand substantial sonication without any apparent degradation in morphology. The desired ATO_3_ materials were then prepared by annealing the particles at high temperature, a process which stimulated the mixing of the ions, which consequently resulted in the formation of the targeted perovskites. Fortunately, the isolated product particles retained the morphology and size of the initial templates.

This surfactant-free procedure we have put forth is advantageous for several reasons. First, it is known that surfactants can bind strongly onto the surfaces of as-prepared particles and are therefore very difficult to completely remove. Because the presence of surfactants can hinder MOR performance [27], any method that precludes their use is catalytically significant. Second, a surfactant-free synthetic protocol is important for rendering the reaction more environmentally friendly. As mentioned, surfactants must be eliminated, because they interfere with optimal electrocatalytic behavior. However, procedures that can effectively remove surfactants are often energy intensive and/or use hazardous chemicals [28,29]. In addition, most surfactants are characterized by long-chain carbons, many of which are relatively harmful to the environment. Even if non-toxic surfactants were to be used, synthesizing these molecules risks creating toxic byproducts [30]. Therefore, the best option would be to do away with surfactants, completely. As such, our synthetic method herein represents an important step towards enabling green chemistry in terms of eliminating the need for surfactants altogether in the generation of ATiO_3_. Third, our 3D motifs are stable in acidic environments and are less likely to aggregate than either carbon supports or clusters of smaller, individualized nanoparticles. In the latter case, the agglomeration of nanoparticles can only be overcome by the use of either surfactants or stabilizing agents [31], which, as we have seen, are detrimental to the observed MOR performance. Fourth, our synthetic route can reliably and reproducibly produce gram-scale quantities of these perovskite supports. In effect, it is a scalable process with potential implications for its applicability in the context of possible commercialization.

To understand the effect of changing the identity of the A-site atom on MOR performance, we assessed a series of 3D ATO micron-scale spheres as supports for Pt nanoparticles. From these data, we noticed several interesting trends. First, with respect to the mass activity, all of the 3D micron-scale spheres of Pt/CaTiO_3_ (CTO), Pt/SrTiO_3_ (STO), and Pt/BaTiO_3_ (BTO) gave rise to enhanced performance, as compared with either a Pt/TiO_2_ commercial particulate standard or Pt/TiO_2_ 3D template controls. Second, for both specific activity and mass activity values, the observed trend for the 3D spheres was in the order of Pt/CTO > Pt/STO > Pt/BTO, with the Pt/CTO system exhibiting the best performance. Third, we measured increased Pt mass activities for our 3D particles of BTO, CTO, and STO supports, even though all of these materials possessed lower exposed surface areas, as compared with the precursor 3D TiO_2_ templates. That is, the inverse correlation between surface area and performance suggests that chemical composition is the most significant parameter in dictating catalytic behavior for perovskites, an assertion consistent with our previous SRO data [19].

## 2. Results and Discussion

### 2.1. General Observations

X-ray powder diffraction (XRD) patterns for all as-prepared micron-scale spheres were obtained and are presented in Figure 1. Experimental data are matched up with relevant known “Joint Committee on Powder Diffraction Standards” (JCPDS) reference database patterns. As-fabricated TiO_2_ intermediate samples consisted of the pure anatase phase. The associated XRD and SEM data are shown in Figure 1A and Figure 2A–C, respectively. Indeed, micron-sized TiO_2_ spheres, with diameters ranging from 3 to 6 microns, were used as templates, to further generate the corresponding ATiO_3_ micron-scale spheres. In particular, the titania template consisted of bulky but porous spherical motifs, characterized by a relatively “smooth” external surface.

These precursor TiO_2_ templates were subsequently converted into their perovskite analogues, in a high-temperature-mediated process. Specifically, our intended targets included not only the desired orthorhombic phase of CTO but also the cubic phase of BTO and STO. In the latter case of STO, we were also able to obtain a pure material with no apparent impurities, such as Sr_2_TiO_4_, by using the hydrothermal method. However, because we could not readily isolate a spherical morphology by using this protocol, we deemed it to be a control sample. These products were isolated after the heating and acid purification processes, as described in the Experimental Synthesis section.

With the BTO samples, the synthesis protocol incorporating a similar high-temperature calcination process ended up generating the anticipated cubic phase of BTO, but this was accompanied by small but detectable quantities of both BaTi_5_O_11_ and barite impurities. In unpublished runs, we found that BaTi_5_O_11_ could be removed and the barite impurities could be minimized but not completely eliminated.

### 2.2. CaTiO_3_

SEM images for the as-synthesized 3D CTO spheres and related size distributions are provided in Figure 2D–F, with isolated products possessing an average measured diameter of 3.7 ± 1.1 μm. The surface appears to be considerably more roughened as compared with the starting TiO_2_ templates. Nonetheless, the average diameters of the starting TiO_2_ templates were 3.6 ± 1.0 μm, implying that there was no substantial change in particle size associated with the chemical transformation from TiO_2_ to CTO.

To further investigate the “phase-transition” process, the as-prepared CTO intermediates were annealed at various temperatures, ranging from 600° to 1100 °C. XRD patterns were collected on a series of purified samples, to observe the nature of the transition from TiO_2_ to the resulting perovskite crystal, as shown in Figure 3. With respect to the growth of CTO, the intermediate gradually transformed into a mixture of TiO_2_ and orthorhombic CTO. The sharp peak located at ~25.46° in the XRD pattern could be ascribed to the anatase phase of TiO_2_. As the temperature increased, the corresponding intensity of the TiO_2_ peak decreased, whereas the signal ascribed to CTO became stronger and more prevalent, a finding indicative of the presumably successful transformation of the TiO_2_ template into the desired perovskite.

Both anatase TiO_2_ and CTO are present at 600°, 700°, and 800 °C, respectively. Beginning at 900 °C, a rutile impurity, situated at 27.52°, appeared and remained until 1100 °C, at which stage, a pure orthorhombic CTO pattern was observed. Appendix A provides for complementary SEM images of the CTO spheres, synthesized at these various different annealing temperatures. There is little change in the morphology as the reaction temperature was increased. Nonetheless, it was observed that the morphology of the relatively smooth and porous TiO_2_ precursor template spheres evolved into a progressively rougher motif, characterized by larger crystallite sizes with rising reaction temperatures. These findings are in line with a corresponding increase in crystallinity, as noted in the associated XRD patterns.

XRD patterns of the corresponding unpurified samples are given in Appendix A. The small peaks located at 29.36° and 31.38° could be assigned to CaCO_3_ impurities. The existence of a TiO_2_ peak is consistent with our assumption that the “sea-urchin” motifs are likely to be TiO_2_ crystals, which can be removed by means of nitric acid purification. An SEM image of the as-described impurities is highlighted in Appendix A.

### 2.3. BaTiO_3_

Similarly, the formation of BTO was observed at different temperatures, ranging from 600° to 1100 °C, with the matching XRD patterns, as shown in Figure 4. Specifically, the cubic phase of BTO initially formed at 600 °C, and its crystallinity appeared to increase as the synthesis temperature was correspondingly increased (Figure 4). No TiO_2_ was observed within any of the XRD patterns of BTO, thereby implying the perovskite may have formed at a lower reaction temperature than what we actually tested. However, the aforementioned BaTi_5_O_11_ and barite impurities emerged at the same time, and could not be properly removed by using either higher temperatures or a nitric acid wash. Appendix A presents XRD patterns of prewashed BTO samples. Because the impurity incorporated excess titanium, we correspondingly increased the amount of Ba(OH)_2_ precursor, to ensure the complete conversion of TiO_2_ to BTO. Though that process could minimize the amount of impurity, it was not possible to fully eliminate all such contaminants in this manner.

Appendix A highlights SEM images of BTO, prepared at different temperatures. As with CTO, the sizes of the individual constituent particles forming the spheres increased with rising reaction temperatures, a finding consistent with the analogous growth in the crystallinity within the XRD patterns. For the BTO samples, the average diameters of the isolated products were determined to be 3.2 ± 1.0 μm, i.e., comparable in size to that of CTO spheres (Figure 2J–L) and consistent with the dimensions of the starting precursor TiO_2_ templates.

### 2.4. SrTiO_3_

STO spheres were synthesized, using three different procedures. The first involved a hydrothermal process described in Section 3.2.3, whereas the second was analogous to the multi-step protocol, using the TiO_2_ template, provided in Section 3.2.2. Finally, in a third series of experiments, ultra-small sub-10 nm samples were generated by comparison, as discussed in Section 3.2.4, using a “water-free” solvothermal procedure. Multiple methods were used to fabricate STO, since the annealing protocol consistently produced impurities within the STO sample. Figure 5 highlights XRD patterns of STO, created at different annealing temperatures. Anatase TiO_2_ can be observed at all such temperatures, including values at and below 900 °C. At 1000 °C, the TiO_2_ is fully transformed to STO; however, a Sr_2_TiO_4_ impurity remains. At 1100 °C, Sr_2_TiO_4_ dominates.

These collective results clearly show that an annealing temperature of 1000 °C is needed to successfully form STO micron-scale spheres, incorporating the smallest amount of impurities. The data shown in SEM Figure 2G–I highlight the isolated morphology and size of representative 1000 °C samples, associated with STO; average diameters were in range of 3.2 ± 1.0 μm. Appendix A illustrates SEM images of STO samples, generated at different annealing temperatures. There is a notable degradation of the overall spherical morphology at 1100 °C, likely due to Sr_2_TiO_4_ formation. As with the CTO sample, as-formed TiO_2_ impurities appeared as rod-like morphologies, which could be successfully removed using a nitric acid wash. Appendix A is an SEM image associated with the pre-nitric acid washed sample.

The panels in Figure 6D–F feature the size histograms and associated morphology of as-prepared STO nanocubes, created with the hydrothermal method. The average diameters of these STO nanocubes were determined to be 130 ± 28 nm. The SEM images of the hydrothermally derived sample suggested that the precursor spheres immediately degraded and broke apart under sonication into either aggregates or individual particles. Hence, even though this synthesis method was able to fabricate pure STO (i.e., XRD data in Appendix A), the isolated product did not retain the desired 3D morphology, especially after washing and addition of Pt. In addition, the sub-10 nm particles of STO were synthesized by using a “water-free” solvothermal reaction. TEM data and the corresponding size histograms of the ultra-small sub-10 nm STO particles can be observed in Figure 6G–I. The solvothermal method used herein generated monodisperse nanoparticles, possessing an average diameter of 4.9 ± 1.0 nm. The associated XRD pattern (Appendix A) suggests that these sub-10 nm STO particles are pure; nevertheless, it was also characterized by the presence of very broad peaks, indicative of not only their small size but also their poor crystallinity. Overall, as neither the STO cubes nor their ultra-small counterparts were optimal in terms of simultaneously achieving monodisperse size and good crystallinity, both of these samples were used as comparative controls with their 3D counterparts in the ensuing MOR tests.

### 2.5. Mechanistic Insights into Structure

The proposed mechanism for the chemical transformation of TiO_2_ to BTO, CTO, and STO, respectively, is illustrated in Figure 7. The first step of the synthesis involves the formation of porous TiO_2_ template spheres, without the use of surfactants. This is achieved by dissolving the titanium butoxide precursor in anhydrous ethanol and sulfuric acid. It has been shown that sulfuric acid is important for promoting the formation of the porous spheres, as it slows the hydrolysis of titanium butoxide to TiO_2_ [32]. This scenario allows for the controlled growth of large and robust spherical particles that are composed of many smaller nanoparticles, in the high temperature and pressure of the autoclave. The second step involves the conversion of TiO_2_ to the desired perovskite spheres. It has been reported that the dried TiO_2_ spheres retain the favorable porous structure mentioned previously [26]. The high degree of porosity implied that precursors of Ba(OH)_2_, Ca(OH)_2_, and Sr(OH)_2_ could adequately and readily diffuse throughout the spherical motifs. The subsequent sonication procedure was used to evenly distribute the Ca^2+^, Ba^2+^, and Sr^2+^ ions throughout the entire motif, thereby allowing for a faster, more homogeneous, and more controlled transformation into the desired perovskite spheres during the ensuing annealing step. It is also worth highlighting the importance of the acid purification protocol, since this was found to remove extraneous, undesired sea-urchin-like, micron-scale flakes present within the sample. It is highly probable that these “sea-urchin” impurities could be ascribed to TiO_2_ precursors, an observation consistent with what has been reported previously [33]. As a general comment, the acid purification procedure assisted in removing other chemical impurities, including not only calcium carbonate formed within the CTO sample but also TiO_2_ “sea-urchins” and rods.

As a means of probing the intrinsic physical porosity of the resulting samples, Table 1 highlights the Brunauer–Emmett–Teller (BET) surface area measurements, collected on the various as-prepared samples. Though commercial TiO_2_ nanoparticles (accompanying XRD provided in Appendix A) were found to sustain a surface area of 9.93 m^2^/g, our 3D TiO_2_ templates possessed the highest surface area of any of the samples tested at 179.22 m^2^/g. Interestingly, despite the consistency in isolated particle sizes, there is a notable decrease in surface area, when the TiO_2_ is subsequently converted into CTO, STO, and BTO, because the corresponding BET surface areas of 2.59, 5.74, and 36.07 m^2^/g, respectively, of these annealed perovskites are much lower than those of the 3D anatase precursor motif. This decrease in surface area can be potentially attributed to the incorporation of the larger Ca^2+^, Sr^2+^, and Ba^2+^ ions within the TiO_2_ lattice, which would have had the apparent effect of reducing pore volume.

Whereas the measured particle diameters derived from SEM data for 3D motifs of TiO_2_, CTO, STO, and BTO were consistently in the range of 3.2 to 3.7 μm, the corresponding crystallite sizes of these materials, computed from the XRD patterns using the Debye–Scherrer equation, yielded a strikingly different trend. Specifically, all of the perovskite samples evinced a consistent increase in calculated crystallite size to >30 nm from the initial 5.67 nm size, associated with the precursor TiO_2_ templates. Our data therefore suggests that this perceptible augmentation in crystallite size for the perovskites is correlated with a reduction in active, available surface area for their reactivity (Table 1). It is also worth noting that our results are consistent with prior literature findings for ATO-type materials, which indicate that cationic incorporation can reduce the effective particle surface area, accessible to reaction [34].

By comparison, STO cubes generated with the hydrothermal method yielded a BET surface area of 74.36 m^2^/g, whereas ultra-small STO samples possessed a surface area of 81.53 m^2^/g. In both examples, the surface area values were higher than those of the analogous 3D STO spheres. Ultra-small STO incorporates both the smallest particle size (~4 to 5 nm, derived from both TEM and XRD data), coupled with the largest surface area of the STO samples synthesized herein. The STO cubes presented an intermediate case, in that its SEM-derived size of ~130 nm was paired with a calculated Debye–Scherrer crystallite size of ~31 nm. As such, the 3D STO spheres appeared to maintain the largest particle sizes and smallest BET surface areas, whereas ultra-small STO gave rise to the exact reverse scenario, namely the smallest particle sizes along with the largest BET surface areas. This apparent inverse correlation between particle size and surface area was consequential for subsequent MOR measurements.

### 2.6. Probing the Effect of Chemical Composition on MOR

To understand the effect of changing the identity of the A site atom on MOR performance, we loaded not only all of the 3D ATO particles but also both standard reference samples and commercial controls in an identical manner with 20 wt% of Pt nanoparticles, using the method described in Section 3.2.5. To confirm the successful deposition of Pt, representative TEM images were obtained and are shown in Appendix A. In addition to microscopy, we collected electrochemical data and analyzed the shape of the associated cyclic voltammogram (CV) curves, which evinced the expected Pt profile. Finally, we also detected a clear color change from white to black of the as-dispersed solution upon the addition of NaBH_4_ during the sample preparation process. All of these observations were collectively consistent with the reduction of the Pt precursor and the corresponding formation of Pt nanoparticles in all samples, prior to MOR data acquisition.

The oxidation of methanol on nanoscale elemental Pt catalysts requires a significant overpotential, due to the formation of partially oxidized intermediate species [35,36]. At low overpotentials, methanol is oxidized via an α-dehydrogenation pathway that leads to the formation of adsorbed carbon monoxide. The buildup of carbon monoxide, commonly referred to as CO poisoning, blocks active sites and leads to poor oxidation kinetics near the thermodynamic potential for MOR. As the overpotential is increased, the surface of Pt is oxidized, and the adsorbed Pt-O* and Pt-OH* groups catalyze the conversion of CO to CO_2_, which reduces the CO coverage and increases MOR kinetics. Thus, the onset of methanol oxidation is typically coincident with the potential wherein surface oxidation occurs.

From a mechanistic perspective, metal oxides play both a passive role, by stabilizing the Pt nanoparticles from aggregation, and a corresponding active role in catalysis, since their surfaces provide oxygen species throughout the entire potential window for methanol oxidation. The presence of oxygen species at the metal–metal oxide interface can facilitate a process referred to as CO spillover, wherein CO species formed on Pt are oxidized by oxygen species present at the NP–support interface. This effect can lead to significantly better CO tolerance. Metal oxides can also influence the electronic properties of adsorbed metal nanostructures through the strong metal–support interaction (SMSI) effect. This effect has been previously shown to increase the activity of gold NPs on titanium dioxide toward CO oxidation in the gas phase [37]. In the context of methanol oxidation, the interaction between Pt NPs and an underlying crystalline RuO_2_ nanostructure resulted in a measurable shift in the oxidation of Pt to lower potentials, which facilitated CO oxidation and thereby lowered the overpotential for methanol oxidation [38].

The different types of perovskite nanoparticles synthesized in this manuscript therefore have enabled us to effectively evaluate the influence of two key parameters, namely support size and composition, upon the activity of immobilized Pt NP catalysts. First, we employed cyclic voltammetry, to investigate the electrochemical properties of the Pt catalysts on the underlying perovskite supports. CV curves for the Pt/CTO, Pt/STO, and Pt/BTO samples are provided in Figure 8. By comparison, Appendix A contains the corresponding CV curves for the Pt/TiO_2_ commercial, Pt/TiO_2_ templates, Pt/STO hydrothermal, and Pt/STO ultra-small samples. All of these data were collected in an Ar-saturated solution of 0.1 M perchloric acid, using a scan rate of 20 mV/s. We note that neither Ti nor the A-site atoms are electrochemically active within this potential window, and therefore are not expected to contribute any faradaic features to the observed CV profile [39,40].

With respect to the structure of the CVs, signals attributable to the reversible hydrogen adsorption (H_ads_) and surface oxidation of the Pt catalyst can be observed in all of the Pt/ATO_3_ samples. The structure of the hydrogen adsorption region (0–0.3 V) is consistent with supported Pt nanoparticles [41]. The oxidation of Pt at potentials above 0.6 V leads to the characteristic oxide region that is observed on the CVs. The onset of surface oxidation is a key parameter for methanol oxidation, since adsorbed oxygen species are necessary to oxidize CO intermediates formed at low overpotentials. A careful analysis of the onset region for the surface oxidation of the Pt NPs reveals that the Pt NPs supported on CTO behave dissimilarly relative to that of the corresponding Pt on analogous STO and BTO supports.

Specifically, the Pt/CTO CV displays three distinctive oxidation waves, beginning at ~0.6, ~0.8, and at 1.0 V. By contrast, the onset for the surface oxidation of Pt within the Pt/BTO and Pt/STO samples does not occur until 0.9–1.0 V. The Pt oxidation features at 0.6 and 0.8 V in the Pt/CTO suggest that there is a unique interaction between the catalyst and the support in CTO, which leads to Pt sites that are more easily oxidized than with either BTO or STO supports. The position of the oxide reduction peak in the cathodic sweep also provides evidence that there is a systematic effect of the A-site cation on the oxygen binding strength. Fits of the reduction peak reveal that there is a shift in the oxide peak position from 0.783 to 0.792 V with increasing cation size. This suggests that there is an inverse correlation between the cation size and the overall binding strength of oxygen on the platinum catalyst’s surface.

In prior reports, these features have been attributed to the presence of active Pt sites located at or near the interface between the Pt NP and the oxide support. These Pt atoms at the interface display unique properties, due either to the SMSI effect and/or to partial coverage of the platinum with an oxide layer at the interface [38]. This unique property of the CTO support is expected to significantly improve CO tolerance, and we believe that this arises because of the trend in composition, which is discussed in detail in the next few paragraphs.

The electrochemically accessible Pt specific surface area (SSA) was calculated from the integration of the hydrogen absorption and desorption peaks in the CV curves. The data related to this analysis are provided in Appendix A. The SSA values for Pt supported on CTO, STO, and BTO are calculated to be 5.80, 3.85, and 5.70 m^2^/g, respectively. These are all significantly higher readings than that of either the TiO_2_ template controls (0.28 m^2^/g) or commercial TiO_2_ nanoparticles (0.60 m^2^/g), which are likely indicative of improved Pt dispersal and immobilization on the perovskite surface. This finding may be due in part to the HNO_3_ washing step. Specifically, pre-treatment with HNO_3_ has been shown to increase both the number and stability of acidic sites on perovskites by exposing Ti on the surface [42,43]. Moreover, these acidic surface sites likely improved the dispersion of the Pt particles during the deposition step and led to higher overall measured SSA values[16]. In addition, the TiO_2_ supports are significantly more porous than the analogous perovskite supports. As such, it is possible that the Pt is deposited more deeply within the porous structure of the TiO_2_ supports and, hence, is not as electrochemically accessible.

Figure 9 highlights the MOR linear sweep voltammetry (LSV) curves and shows the expanded onset regions, with respect to Pt mass activity (Figure 9A,B) and specific activity (Figure 9C,D), for the Pt/CTO, Pt/STO, and Pt/BTO samples. Figure 10 presents bar graphs, showing the comparative MOR performance for the mass and specific activity at 0.7 and 0.8 V. Appendix A includes a summary of results, derived from MOR measurements obtained at both 0.7 and 0.8 V. From these data, we noticed several interesting trends within the measurements. First, with respect to the mass activity, all of the 3D micron-scale spheres of Pt/CTO, Pt/STO, and Pt/BTO evinced enhanced performance, as compared with either a Pt/TiO_2_ commercial particulate standard or Pt/TiO_2_ 3D templates. Second, for both SA and MA values, the activity trend for the 3D spheres was in the order of Pt/CTO > Pt/STO > Pt/BTO, with CTO exhibiting the best performance. This was consistent with our observations of an early onset of Pt oxide formation in the Pt/CTO sample from the CV data.

In order to rationalize the trend in performance, we considered several plausible explanations that could lead to an increase in activity in the CTO and STO samples relative to that of BTO. For example, we correlated these behaviors with parameters such as (i) BET surface area; (ii) the measured particle size obtained from SEM and TEM; and (iii) the crystallite size, which is calculated by using the Debye–Scherrer equation from the XRD patterns, all of which are provided in Table 1. Interestingly and somewhat counter-intuitively, we measured increased mass activities for our 3D particles of BTO, CTO, and STO, even though all of these materials possessed lower BET surface areas as compared with the precursor 3D TiO_2_ templates. Though it would have been expected that a higher surface area of the supports would yield a correspondingly greater catalytic performance, due to the increase in the number of exposed active sites [19], we observed the exact opposite herein. That is, the inverse correlation between surface area and performance suggests that chemical composition is a more significant parameter as opposed to the support surface area and particle size in dictating catalytic behavior for perovskites, an assertion consistent with our previous findings with SRO [19].

As an additional contributing factor to activity enhancement for ATO systems, we point out that all of the perovskite samples, in addition to the commercial TiO_2_ nanoparticles, maintained calculated Debye–Scherrer crystallite sizes of over 30 nm. This result represents a significant increase as compared with the 5.7 nm associated with the TiO_2_ precursor template, implying a greater degree of crystallinity for the perovskite products. The exact effects of crystallinity on catalytic performance are complex, with some reports showing that the use of amorphous materials resulted in higher performance metrics [44,45]. By contrast, other studies implied that an increase in crystallinity of the support correlated with improved MOR performance [46]. In a prior report, well-defined crystalline octahedral SrRuO_3_ NPs were found to yield higher MOR activity than spherical nanoparticles with disordered surfaces [13,19]. In this case, the higher degree of crystallinity may also contribute to the overall higher activity of the perovskite particles relative to that of TiO_2_. However, the crystallinity of all of the perovskite supports we synthesized is similar, and thus, this factor alone does not fully explain the observed trend in performance within the set of perovskite supports tested. 

Within the series of titanate-based perovskites themselves, CTO yielded the best performance, even though all of the ATO spheres possessed similar particle sizes and crystallite sizes, as deduced from electron microscopy and diffraction data. Moreover, all samples were identically processed in terms of not only annealing temperatures used but also the relevant acid washing steps. Hence, to gain insights into this behavior, we postulate that this activity trend correlates with the ionic radius of the “A” site cation and the surface energy of the corresponding perovskite crystal lattice. Specifically, because Ca^2+^ < Sr^2+^ < Ba^2+^ in terms of ionic radius, CTO possesses the lowest surface energy, the weakest water binding, and the least exothermic enthalpy of formation as compared with these alkaline-earth perovskite analogues [47]. Such factors correlate with the observed trend in MOR performance of Pt/CTO > Pt/STO > Pt/BTO, highlighting the collective importance of ionic radius, surface energy, and formation enthalpy in accounting for CTO’s higher catalytic activity as compared with STO and BTO [47,48].

Although it is difficult to ascribe a single explanation for the influence of the A-site cation upon performance, there are several plausible scenarios, based on prior literature, that are related to the structure and electronic properties of CTO, BTO, and STO. For example, it is evident from the CV data that CTO leads to the oxidation of Pt surface sites at significantly lower potentials. One plausible rationale for this observation is the low hydration energy of CTO in relation to the larger cations. A lower hydration energy may reduce the coverage of water on the surface of CTO, thereby leading to a stronger interaction of Pt sites at the interface between the Pt NP and the metal oxide substrate. Alternatively, prior literature has shown that CTO favors the exposure of the {001} surface facets due to a subsurface reconstruction, which is not favored in either STO and BTO [49]. Variations in the surface structure of perovskites can lead to significant impacts upon surface energy, oxygen coverage, and also the degree of surface rumpling, all of which can contribute to a unique interaction between the CTO surface and Pt in addition to the predicted trend in oxygen binding strength on the Pt catalysts.

To further understand the performance of our 3D ATO microspheres, we collected chronoamperometry results, which are shown in Figure 11. These tests were run for a period of 3600 s, at a constant voltage of 0.8 V vs. RHE, a typical value within the range of methanol oxidation. Chronoamperometry data for the standards and templates are provided in Appendix A. There is an initial activity drop observed for all samples that may be due not only to the buildup of poisoning intermediate species (such as CO) but also to the small degree of surface oxidation that occurs at this potential. These species would poison the active sites, thereby resulting in a perceptibly diminished signal upon the initiation of methanol oxidation [50]. After the rapid initial drop, a steady state was achieved and maintained for all samples.

Pt/CTO evinced the best performance, with Pt/BTO and Pt/STO possessing very similar steady-state activities. This finding suggests that the CTO substrate aids in the oxidation of CO to CO_2_ and reduces Pt poisoning, as compared with the other samples analyzed. Moreover, the reference standards tested, including the TiO_2_ commercial, TiO_2_ templates, STO ultra-small, and STO hydrothermal specimens, exhibited substantially lower steady-state activities, as compared with their 3D ATO micron-sphere analogues. Hence, these data are consistent with what we have identified previously, namely that the small ionic radius of Ca^2+^ results in CTO yielding the highest steady-state current densities. Not surprisingly, our 3D ATO micron-scale spheres significantly outperform both TiO_2_ and STO samples characterized by different morphologies.

### 2.7. Probing the Effect of Size and Morphology on MOR: STO Series

As mentioned earlier, we generated a series of STO samples, in the form of spheres, nanocubes, and ultra-small particles possessing distinctive sizes and morphologies, created with different surfactant-free synthesis procedures. The annealed 3D STO sample gave rise to the highest relative specific and mass activity values, as compared with those derived from either hydrothermal or solvothermal methods. As we observed previously, these performance data inversely correlated with BET surface area trends, in that the readings associated with STO ultra-small, STO cubes, and 3D STO spheres were 81.52, 74.36, and 5.74 m^2^/g, respectively, implying, in general, that surface area was not an important factor.

With regards to ultra-small STO nanoparticles, their poor performance may be explained by a combination of their amorphous nature coupled with their small crystallite size, 4.3 ± 0.4 nm, as determined by XRD [46]. Because the size of the ultra-small STO particles is very close to that of the associated Pt nanoparticles (i.e., ~2 nm), the deposition step likely resulted in a physical mixture of both STO nanoparticles and Pt nanoparticles (Appendix A), as opposed to a more clearly differentiable catalyst/support architecture that would have been predicted to form in the case of larger STO particles. Interestingly, 3D STO displays higher catalytic activity relative to that of analogous STO cubes, despite both samples possessing comparable crystallite sizes. Although it is difficult to precisely identify the origin of this trend, the 3D STO particles were subjected to an acid treatment, whereas STO cubes were prepared under basic conditions. As discussed earlier, in acid, the surfaces of the 3D STO particles would be expected to preferentially expose Ti surface sites. As such, with 3D STO, the presence of such surface sites may lead to not only a higher overall stability during electrocatalysis in acidic media but also a desirable decrease in potentially deleterious Pt aggregation [16].

## 3. Materials and Methods

### 3.1. Materials

All chemicals were used as received, without further purification. Specifically, the metal precursors of strontium chloride (SrCl_2_ 99–100.4%), barium hydroxide hexahydrate (Ba(OH)_2_ 6H_2_O, 98.0%), and calcium hydroxide (Ca(OH)_2_, 98%) were purchased from either J.T. Baker or Baker & Adamson Chemical. Sodium hydroxide pellets (NaOH, 98.5%), sulfuric acid (H_2_SO_4_, 95–98%), strontium metal (99%), and anhydrous benzyl alcohol (99.8%) were acquired from Millipore Sigma. Titanium (IV) n-butoxide (Ti(OBu)_4_, 99%) and titanium (IV) isopropoxide (Ti(C_3_H_7_O)_4_, 98%) were procured from Acros Organics, while dihydrogen hexachloroplatinate (IV) hydrate (H_2_PtCl_6_·x H_2_O, 99.9%), strontium hydroxide octahydrate (Sr(OH)_2_·8 H_2_O, 99%), and an aluminum oxide polishing compound (Al_2_O_3_) were obtained from Alfa Aesar. Anhydrous ethanol (90%) solvent was bought from BeanTown Chemical.

### 3.2. Synthesis

#### 3.2.1. Synthesis of TiO_2_ Templates

The TiO_2_ micron-scale spheres were fabricated by a hydrothermal procedure, modified from a previously reported alcohol-based method [26]. Specifically, 5.1 g titanium butoxide (Ti(OBu)_4_) was placed in a 250 mL round-bottom flask to which 75 mL anhydrous ethanol was added dropwise into the container, with gentle magnetic stirring. It was observed that, after about 20 mL of ethanol had been added, the solution became cloudy white in hue. The mixture was then allowed to stir for 30 min, before 0.33 mL of concentrated sulfuric acid was injected into the beaker. The solution turned clear after several minutes. The whole solution was then kept stirring, at room temperature, for another 2 h. The reagents were subsequently transferred into a 100 mL Teflon autoclave, which was heated to 180 °C for 8 h. The resulting white product produced from the hydrothermal reaction was collected by means of centrifugation and washed with both de-ionized (DI) water and ethanol.

#### 3.2.2. Synthesis of CaTiO_3_, BaTiO_3_, and SrTiO_3_

The desired CTO, BTO, and STO micron-scale spheres were generated by a modified sol–gel calcination process [26]. Specifically, as-prepared TiO_2_ powder templates were dispersed into anhydrous ethanol, followed by adding in either Ca(OH)_2_, Ba(OH)_2_·6 H_2_O, or Sr(OH)_2_·8 H_2_O precursors, respectively, depending on the targeted product. The molar ratio of TiO_2_ to metal hydroxide was fixed at 1:1. In typical runs, we added in 100 mg of TiO_2_ with either 0.092, 0.395, or 0.321 g of Ca(OH)_2_, Ba(OH)_2_·6 H_2_O, or Sr(OH)_2_·8 H_2_O, respectively. The amount of anhydrous ethanol put into the mixture ranged between 15 and 20 mL; however, the exact amount did not appear to make a perceptible difference. The whole mixture was sonicated for 2 h at room temperature.

The resulting solid intermediate species was collected by centrifugation and then dried at 60 °C, in the drying oven, for 1 h. The as-prepared intermediate was calcined immediately after drying, at 1000 °C for 3 h, using a ramp rate of 8 °C/min. Samples were subsequently purified by a combination of centrifugation/sonication (9000 rpm for 5 min) steps in the presence of dilute nitric acid (0.05 M), to remove impurities, as well as residual TiO_2_. The washing procedure involved rinsing 3× with nitric acid, 2× with water, and 1× with an ethanol, in that precise, sequential order, so as to avoid contact between ethanol and nitric acid.

#### 3.2.3. Hydrothermal Synthesis of SrTiO_3_

STO nanocubes were obtained by reacting and converting the precursor TiO_2_ micron-scale spheres, prepared in the first step, by using NaOH as a “catalyst” [51]. In a typical experiment, 0.5 g TiO_2_ spheres in the form of dry powder coupled with 1.67 g strontium chloride hexahydrate powder were weighed and dissolved in 50 mL DI water, under sonication, for 10 min; 2 g of NaOH was then added into the mixture, with magnetic stirring. Once the NaOH had completely dissolved, the mixture was transferred to a 100 mL Teflon-lined autoclave and heated to 140 °C for 4 h. The final product was collected by centrifugation and washed with both DI water and ethanol multiple times.

#### 3.2.4. Ultra-Small SrTiO_3_

Ultra-small, sub-10 nm STO particles were synthesized, using a previously published method [21]. In short, 2 mmol (175.2 mg) Sr metal and 2 mmol (0.591 mL) titanium (IV) isopropoxide were added to 25 mL of anhydrous benzyl alcohol, under an Ar atmosphere, within a glove box, to prevent H_2_O and O_2_ contamination. The mixture was stirred for several minutes and then transferred to a 50 mL Teflon lined autoclave. The autoclave was subsequently sealed and removed from the glove box, prior to heating at 200 °C, for 48 h. The formation of a milky white suspension was observed, and the precipitated samples were washed several times with ethanol, coupled with centrifugation for 10 min at a rate of 9000 rpm.

#### 3.2.5. Depositing Pt Nanoparticles onto ATiO_3_ Micron-Scale Spheres

Pt nanoparticles (20% weight loading) were supported onto ATiO_3_ micron-scale spheres via an in situ growth method [52]. Specifically, ATO spheres were dispersed in 100 mL DI water, to yield a final concentration of 1 mg/mL. In a typical procedure, 42 mg of chloroplatinic acid hexahydrate was introduced to this solution, to give rise to a 20 wt% Pt loading. The solution was then sonicated for 30 min. Next, 50 mL of sodium borohydride solution, incorporating a NaBH_4_ concentration of 2 mg/mL, was then quickly added to the mixture, as a reducing agent, under magnetic stirring. The reaction was further stirred for 2 h. After that last step, the resulting final product, comprising a Pt-loaded metal oxide support sample, was subsequently washed with deionized water and ethanol, and ultimately dried overnight, at 60 °C, in a drying oven. The samples were then re-dispersed in ethanol, with a final concentration of 10 mg/mL.

### 3.3. Structural and Morphological Characterization

#### 3.3.1. X-Ray Diffraction (XRD)

Diffraction experiments were conducted, using a Scintag diffractometer, operating in the Bragg–Brentano configuration with Cu Kα1 irradiation (λ = 1.54 Å). All diffraction patterns of ATO samples were collected with a scanning rate of 10 degrees per minute. Powder samples were dispersed in ethanol and drop-cast onto a zero-background holder (MTI Corporation, zero diffraction plate for XRD, B-doped, p-type Si, 23.6 mm in diameter and 2 mm in thickness), followed by drying in air.

#### 3.3.2. Electron Microscopy

As-prepared ethanolic sample solutions for scanning electron microscopy (SEM) were deposited onto silicon wafers, individually, and then evaporated in air. Initial characterization experiments to deduce morphology were performed on a JEOL 7600 microscope, with the resulting images collected under an accelerating voltage of 5 kV.

For higher-resolution, complementary transmission electron microscopy (TEM), as-prepared ethanolic sample solutions were drop cast onto 3 mm lacey carbon-coated copper grids, prior to analysis. A JEOL 1400 TEM instrument was used to image all of the samples, at an accelerating voltage of 120 kV.

#### 3.3.3. BET Surface Area

Surface area measurements were obtained, using a Nova 2200e surface area analyzer. All samples were degassed at 250 °C, for 2 h, under vacuum, before measurements were taken. A Quantachrome Instruments–derived surface area reference material (Cat. No. 2001), possessing a surface area of 14.26 ± 1.22 m^2^/g (95% reproducibility), was used to calibrate the instrument.

#### 3.3.4. Electrochemical Measurements

Prior to electrochemical characterization, samples of Pt NPs supported onto ATiO_3_ micron-scale spheres were rendered into catalyst inks by dispersing the dry powders into ethanol, so as to create an ~10 mg/mL solution. A glassy carbon rotating disk electrode (GC-RDE, Pine Instruments, 5 mm) was polished, using an aluminum oxide powder (average particle size of 0.3 μm). Two 5 μL drops of the as-prepared catalyst ink were then loaded onto the glassy carbon electrode and air-dried. One 5 μL drop of an ethanolic 0.025% Nafion solution was sealed in the catalyst.

A three-electrode electrochemical cell was assembled with a Pt counter electrode coupled with an Ag/AgCl/3M Cl^−^(aq) electrode, serving as the reference electrode. Electrochemical measurements of Pt NP/ATiO_3_ spherical composites were subsequently performed within a 0.1 M perchloric acid (Optima grade) solution electrolyte, generated from high-purity type 1 water (18.2 MΩ^•^cm). Cyclic voltammogram (CV) curves were collected within an Ar-saturated electrolyte solution, at a scan rate of 20 mV/s. The MOR measurements themselves of the various samples were conducted by obtaining linear sweep voltammograms at a scan rate of 20 mV/s within a de-oxygenated 0.5 M methanol (Optima grade) solution, supported within a 0.1 M perchloric acid electrolytic environment. Chronoamperometry measurements were obtained within the same 0.1 M perchloric acid electrolyte in the presence of 0.5 M methanol. Tests were run at a constant voltage of 0.8 V vs. RHE for 3600 s.

## 4. Conclusions

In this study, we successfully synthesized and characterized 3D spherical motifs of perovskite metal oxides, such as CTO, STO, and BTO. The synthesis method we have developed herein involves a two-step, surfactant-free protocol in which porous 3D anatase TiO_2_ templates are initially synthesized hydrothermally and subsequently reacted with A(OH)_2_ precursors under high-temperature annealing conditions. Based on XRD data collection as a function of temperature, the conversion of TiO_2_ to ATO at elevated temperatures required 1000 °C for most of the samples to fully convert them into the desired perovskite spherical motifs. This conversion process was aided by the porous nature of the TiO_2_ template spheres, allowing for a reasonable distribution of the “A” site ions through sonication, prior to annealing. This protocol yielded spheres that were robust enough to maintain their shape, even under sonication, unlike hydrothermally generated STO, for instance. In addition, this procedure could produce gram-scale quantities of the desired perovskites with relatively clean outer surfaces; the absence of any potentially deactivating surfactants was conducive to favorable electrochemical performance.

In terms of MOR behavior, the CTO sample yielded the highest performance for a number of reasons. Among the perovskites we tested, CTO possessed the lowest surface energy, the weakest water binding, and the least exothermic enthalpy of formation as compared with its alkaline-earth perovskite analogues. The trend in ionic radius of Ca^2+^ < Sr^2+^ < Ba^2+^ correlated very well with the observed trend in MOR performance of Pt/CTO > Pt/STO > Pt/BTO. Moreover, the greater stability and crystallinity of this sample were consistent with our findings that chemical composition, as opposed to either size, morphology, or BET surface area alone, was the most important determinant of MOR activity, corroborating our prior results on SRO.

## Figures and Tables

**Figure 1 molecules-26-00909-f001:**
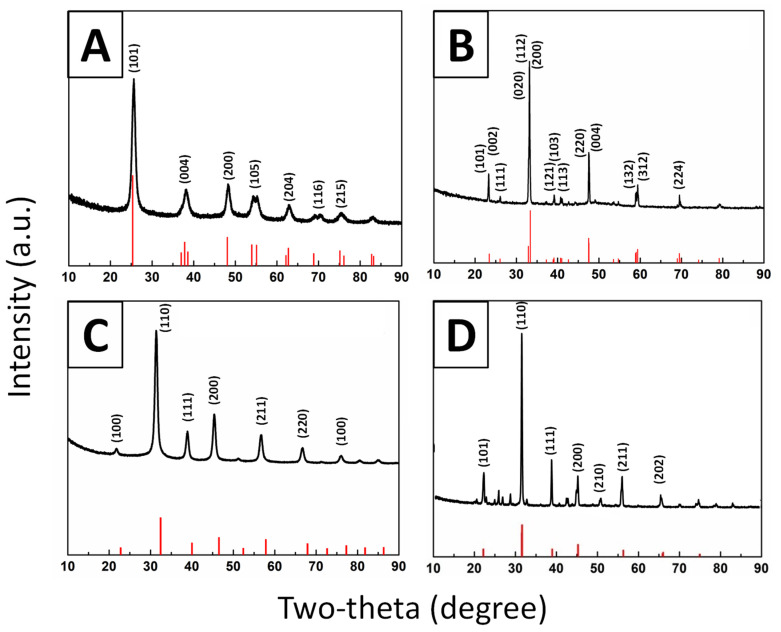
Indexed XRD patterns of (**A**) TiO_2_ intermediate JCPDS #84-1286), (**B**) CaTiO_3_ (CTO) sample (JCPDS #86-1393), (**C**) SrTiO_3_ (STO) sample (JCPDS # 86-0179), and (**D**) BaTiO_3_ (BTO) sample (JCPDS #74-1968) with a small BaTi_5_O_11_ impurity.

**Figure 2 molecules-26-00909-f002:**
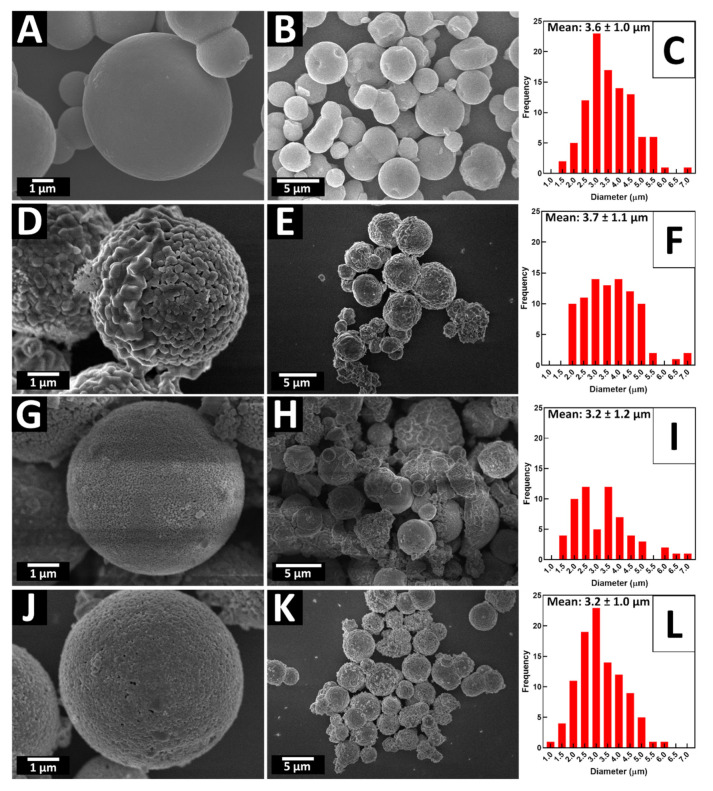
SEM images of ATO samples with associated particle size histograms for (**A**–**C**) TiO_2_, (**D**–**F**) CTO, (**G**–**I**) STO, and (**J**–**L**) BTO micron-scale spheres, respectively.

**Figure 3 molecules-26-00909-f003:**
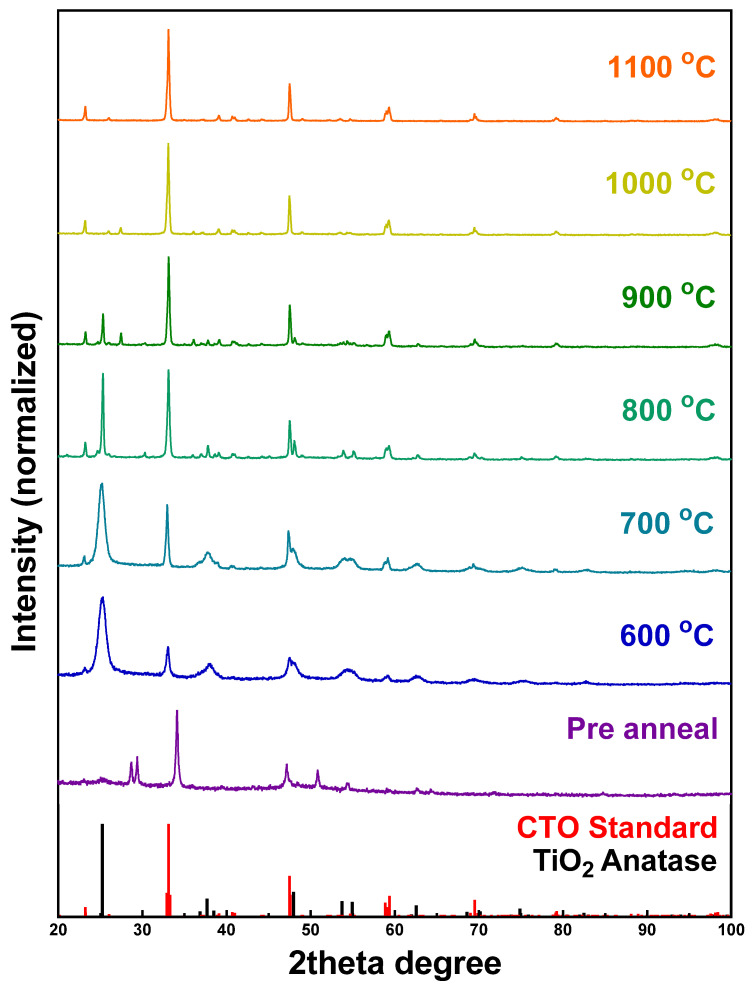
XRD patterns of a CTO powder sample calcined from 600 °C to 1100 °C, in increments of 100 °C. The data on all samples were acquired after the acid wash, except for the pre-annealed sample. Samples are compared with standard anatase TiO_2_ and CTO patterns.

**Figure 4 molecules-26-00909-f004:**
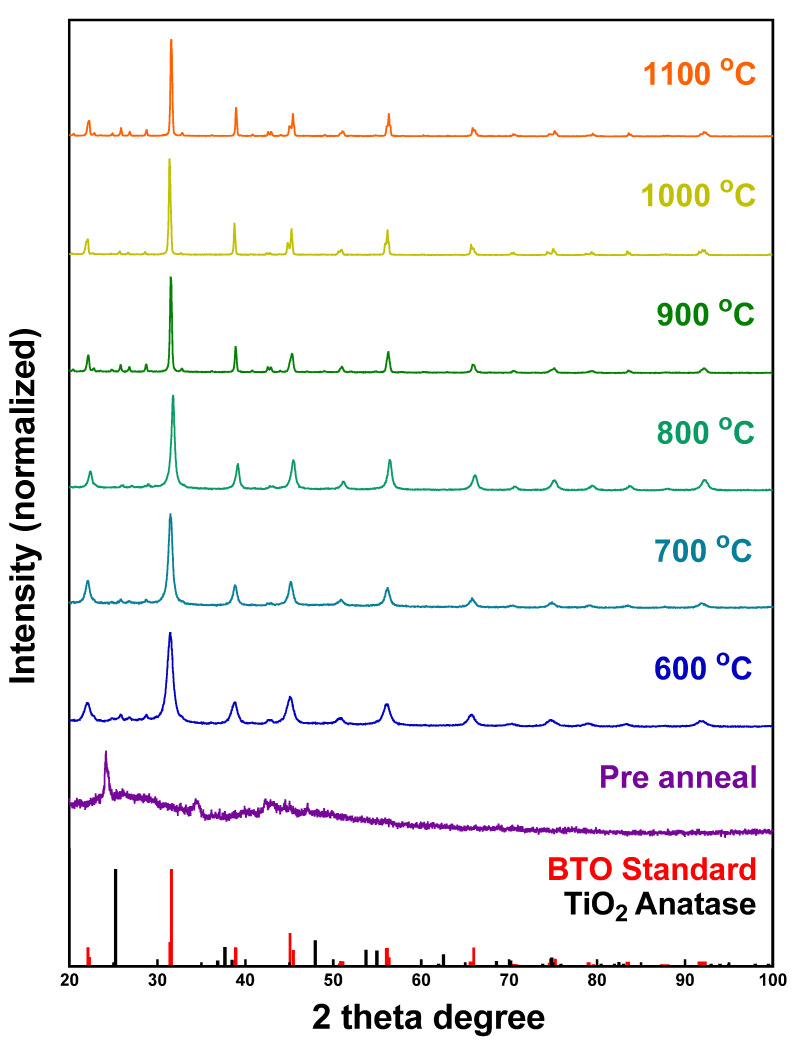
XRD patterns of a BTO powder sample calcined from 600 °C to 1100 °C, in increments of 100 °C. The data on all samples were acquired after the acid wash, except for the pre-annealed sample. Samples are compared with standard anatase TiO_2_ and BTO patterns.

**Figure 5 molecules-26-00909-f005:**
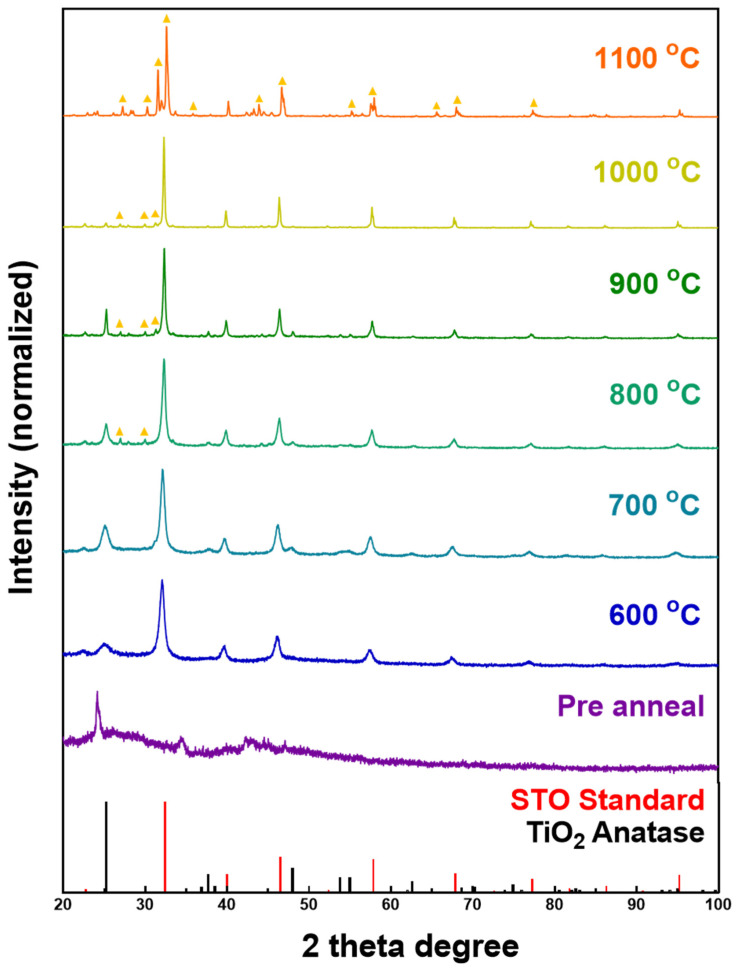
XRD patterns of a STO powder sample calcined from 600 °C to 1100 °C, in increments of 100 °C. The data on all samples were acquired after the acid wash, except for the pre-annealed sample. Samples are compared with standard anatase TiO_2_ and STO patterns. Triangles associated with the 800 °C to 1100 °C samples indicate the presence of Sr_2_TiO_4_ peaks.

**Figure 6 molecules-26-00909-f006:**
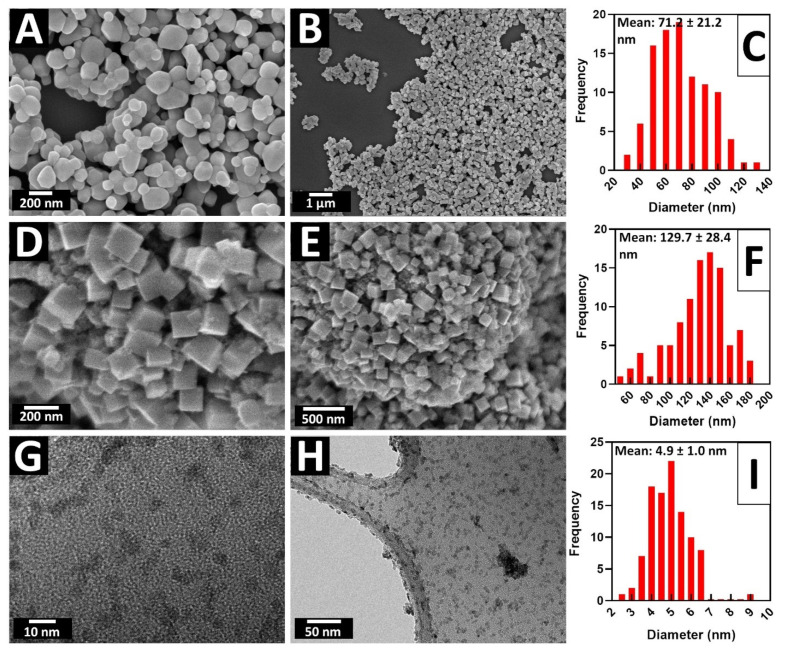
SEM images and associated particle size histograms of (**A**–**C**) commercial anatase TiO_2_ and (**D**–**F**) STO hydrothermally derived nanocubes, along with TEM images and corresponding size histograms of (**G**–**I**) STO ultra-small nanoparticles.

**Figure 7 molecules-26-00909-f007:**
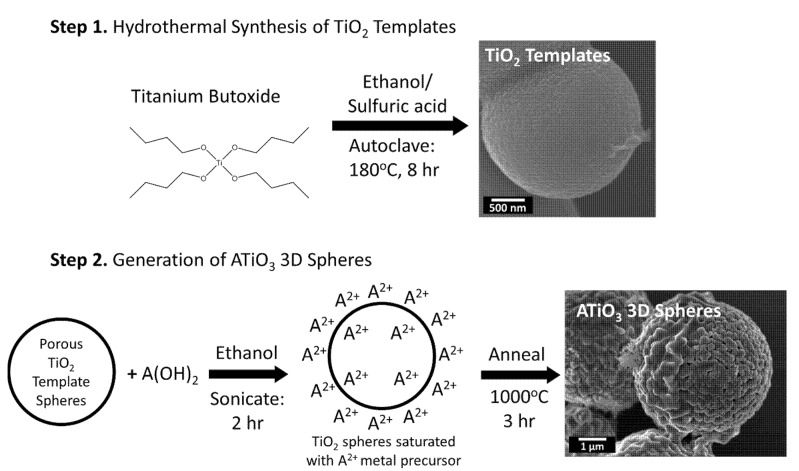
Proposed mechanistic flowchart, associated with the multi-step, surfactant-free synthesis of the as-prepared 3D ATiO_3_ micron-scale spheres.

**Figure 8 molecules-26-00909-f008:**
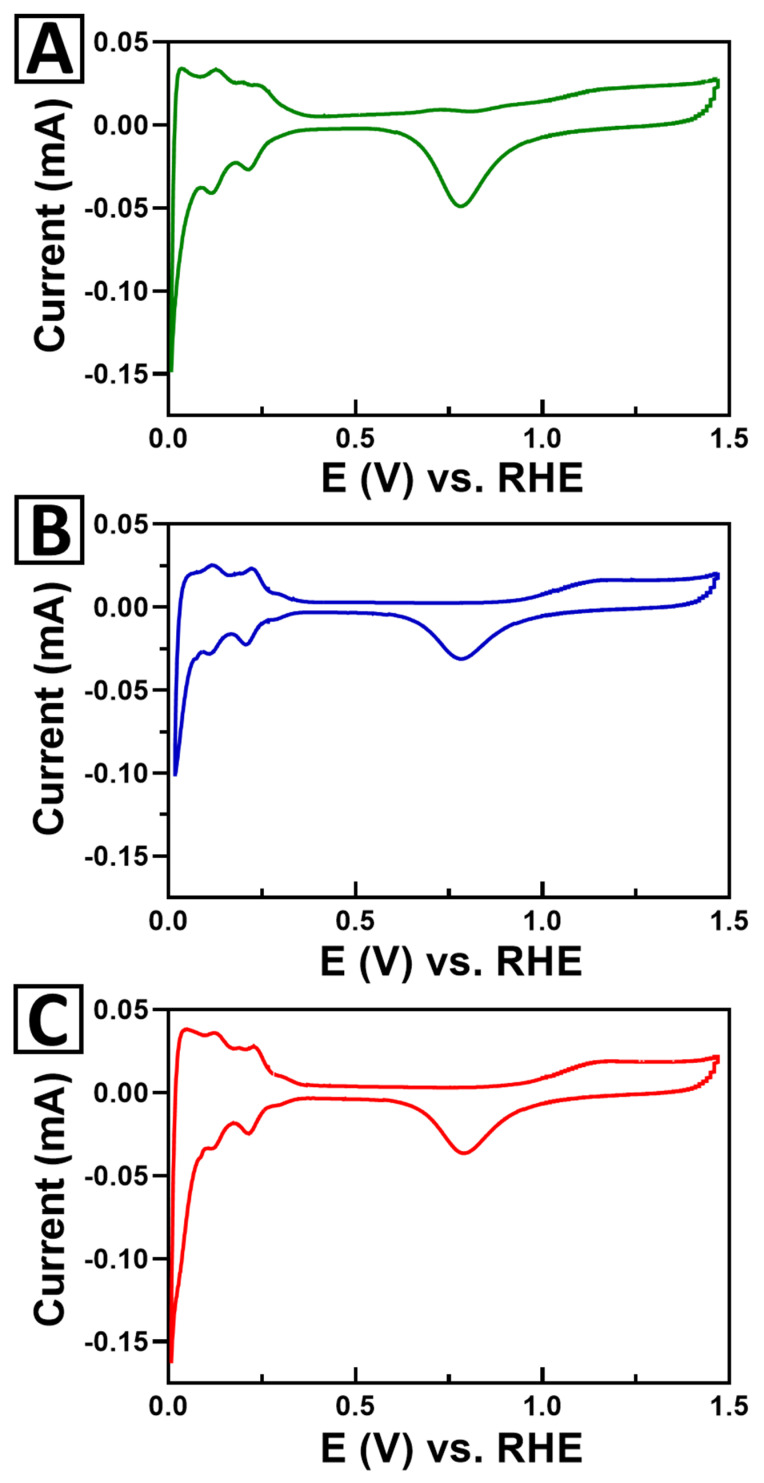
Cyclic voltammogram (CV) curves for 3D spheres of (**A**) Pt/CTO, (**B**) Pt/STO, and (**C**) Pt/BTO, respectively.

**Figure 9 molecules-26-00909-f009:**
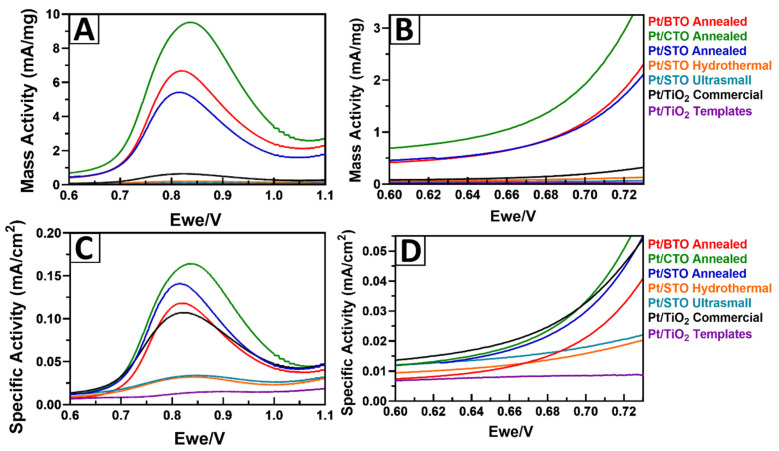
Methanol oxidation reaction (MOR) LSV curves for all samples obtained, at a scan rate of 20 mV/s, within an Ar-saturated 0.1 M perchloric acid solution + 0.5 M MeOH medium. Data highlight (**A**) mass activity, (**B**) the corresponding MA onset region, (**C**) specific activity, and (**D**) the corresponding SA onset region, respectively.

**Figure 10 molecules-26-00909-f010:**
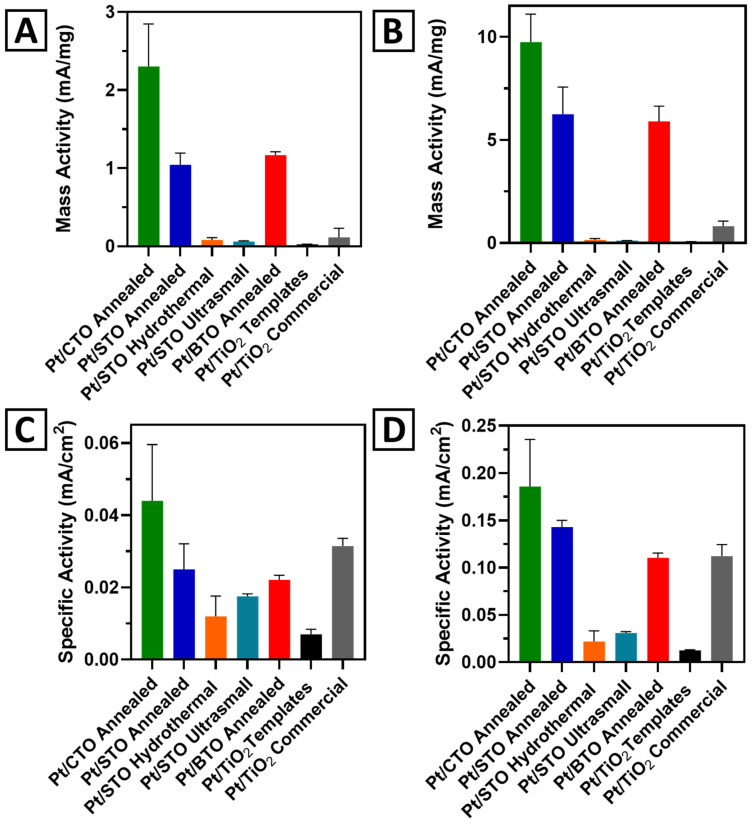
Bar graphs of (**A**) MA at 0.7 V, (**B**) MA at 0.8 V. (**C**) SA at 0.7 V, and (**D**) SA at 0.8 V, respectively, for all ATO samples and standards.

**Figure 11 molecules-26-00909-f011:**
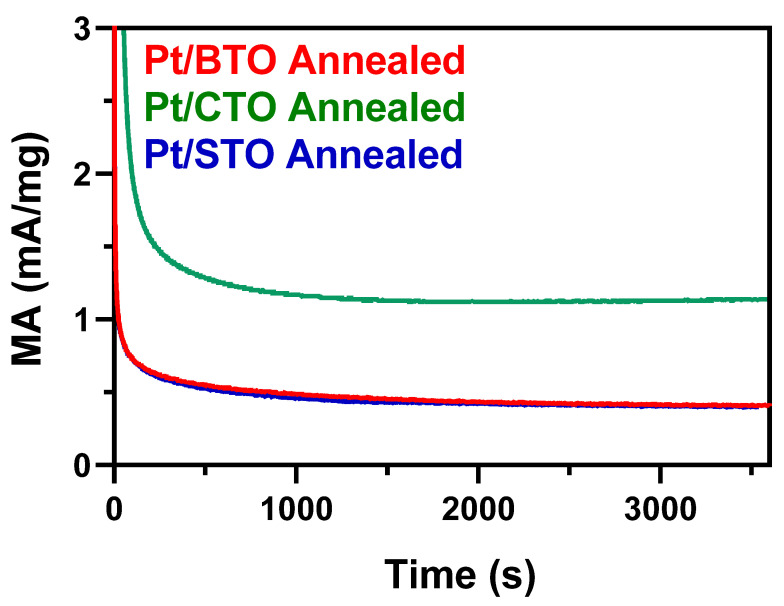
Chronoamperometry measurements at 0.8 V of 3D ATO microspheres obtained within an Ar-saturated 0.1 M perchloric acid solution + 0.5 M MeOH medium for 3600 s.

**Table 1 molecules-26-00909-t001:** BET surface area measurements, microscopy-derived particle sizes, and calculated Debye–Scherrer crystallite sizes of ATO samples, TiO_2_ templates, and commercial TiO_2_.

Sample	BET Surface Area (m^2^/g)	Particle Size (Derived from Either SEM or TEM Measurements)	Crystallite Size from Debye–Scherrer Equation (Calculated from XRD)
TiO_2_ templates	179.22	3.6 ± 1.0 μm	5.7 ± 0.7 nm
STO— ultra-small	81.53	4.9 ± 1.0 nm	4.3 ± 0.4 nm
STO— Hydro- thermal	74.36	130 ± 28 nm	31.1 ± 3.4 nm
BTO— annealed	36.07	3.2 ± 1.0 μm	34.4 ± 8.1 nm
TiO_2_— commercial	9.93	71.1 ± 21.2 nm	39.9 ± 6.3 nm
STO— annealed	5.74	3.2 ± 1.2 μm	32.1 ± 2.6 nm
CTO— annealed	2.59	3.7 ± 1.1 μm	36.0 ± 8.0 nm

## Data Availability

The data presented in this study are available within the current article and accompanying Appendix A section.

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
