# Peer review of "Surfactant-Free Synthesis of Three-Dimensional Perovskite Titania-Based Micron-Scale Motifs Used as Catalytic Supports for the Methanol Oxidation Reaction"

_molecules, 2021, doi:10.3390/molecules26040909_

Round 1
Reviewer 1 Report
The paper is a continuation of groups previous research on MOR over perovskites. The authors show an advantageous, surfactant-free synthesis method of TiO2-based perovskites that is likely to overcome the most important issues occurring in perovskites obtained in a conventional way in the presence of surfactants.
The paper is very well written and points out all of important features of those materials in terms of their physico-chemical properties and performance in MOR. Presentation of the results is also very good and clear.
The paper could by published in the present form; however, I suggest considering a minor revision of its structure:
The introduction part is quite long but it clearly demonstrates the background for the presented research. It also shows some conclusions of the presented study.
For the reader's convenience, I suggest to add a Conclusion section at the end of the document that will sum up the research.
Author Response
Comment: “1. For the reader's convenience, I suggest to add a Conclusion section at the end of the document that will sum up the research.”
Response: The reviewer has asked for us to add a conclusion section at the end of the paper to clarify the results. In response, we would like to respectfully point out that we had already previously included a conclusion section, labeled “4. Conclusion” located on pages 22-23 of the revised version of the paper, which summarized the findings of the current report.
Reviewer 2 Report
The authors synthesized and rationalized the formation of a series of 3D hierarchical metal oxide spherical motifs. A two-step, surfactant-free syn-thesis procedure was used to generate ATiO3 (wherein ‘A’ = Ca, Sr, and Ba) perovskites with average diameters of ~3 microns. Their use as supports for the methanol oxidation reaction (MOR) as a function of their size, morphology and chemical composition are discussed in detail. The study in this article is interesting and helpful for understanding the role of ABO3 perovskite for MOR. I recommend it for publication after some modification.
- the peaks regarding the secondary phases, e.g. BaTi5O11 or Sr2TiO4, are unclear. It is suggested that all the secondary phases are identified by symbols.
- In Figure s4, it is incorrect when only one star symbol is marked for BTO. Please check all the XRD patterns to make sure the display is correct and clear.
Reviewer 3 Report
This paper reported perovskite metal oxides and studied their methanol oxidation reaction behavior. The as prepared materials are very stable even under sonication. This paper is well written with detailed characterizations. The information provide in this paper is valuable for the fields. Overall, I will suggest accepting this manuscript in current form.
Author Response
Response: Reviewer 3 did not request any apparent corrections. We thank for reviewer for the kind comments expressed.
Reviewer 4 Report
The authors studied the effect of changing the identity of the A site atom on MOR performance by assessing a series of 3D ATO micron-scale spheres as supports for Pt nanoparticles. The author has studied in detail the influencing factors of MOR behavior of ATiO3 perovskites from many aspects. The research content is very detailed and credible. I think this work can be accepted directly after the following two small problems are solved:
- Please explain what are the small particles in Fig. 6 G and H?
- In Fig. S9E, it seems that the attached Pt particles and STO ultra-small are not clearly distinguishable from each other.
Author Response
Comment: “1. Please explain what are the small particles in Fig. 6 G and H?
Response: The small particles correspond to the sub-10 nm STO generated through the water-free solvothermal method. To clarify the contents of Figure 6G and H in the text, we have revised the lines located in page 11, lines ~3-8: “TEM data and the corresponding size histograms of the ultra-small, sub-10 nm STO particles can be observed in Figure 6G-I. The solvothermal method used herein generated monodisperse nanoparticles, possessing an average diameter of 4.9 ± 1.0 nm. The associated XRD pattern (Figure S7) suggests that these sub-10 nm STO particles are pure; nevertheless, it was also characterized by the presence of very broad peaks, indicative of not only their small size but also their poor crystallinity.”
Comment: “2. In Fig. S9E, it seems that the attached Pt particles and STO ultra-small are not clearly distinguishable from each other.
Response: Due to the similar size of the Pt particles and the STO ultrasmall particles. it is difficult to tell them apart in the TEM. However, we can confirm the presence of platinum because the shape of the CV curve of the ultra-small STO sample matches that of Pt. Pt was also confirmed to be present, due to the fact that the as-dispersed solution altered from white to black upon the addition of NaBH4 and corresponding reduction of Pt4+ to Pt0. This observation was consistent for all samples. To clarify this in the text, an edit was made to page 13, lines ~14-17: “In addition to microscopy, we also collected electrochemical data and analyzed the shape of the associated CV curves, which evinced the expected Pt profile. Finally, we also detected a clear color change from white to black of the as-dispersed solution upon the addition of NaBH4 during the sample preparation process. All of these observations were collectively consistent with the reduction of the Pt precursor and the corresponding formation of Pt nanoparticles in all samples, prior to MOR data acquisition.”